# Investigation of Combined Cyclodextrin and Hydrogel Formulation for Ocular Delivery of Dexamethasone Acetate by Means of Experimental Designs

**DOI:** 10.3390/pharmaceutics10040249

**Published:** 2018-12-01

**Authors:** Roseline Mazet, Luc Choisnard, Delphine Levilly, Denis Wouessidjewe, Annabelle Gèze

**Affiliations:** 1Faculty of Pharmacy, University of Grenoble Alpes, DPM, UMR CNRS 5063, ICMG FR 2607, F-38400 Saint Martin d’Hères, France; rmazet@chu-grenoble.fr (R.M.); luc.choisnard@univ-grenoble-alpes.fr (L.C.); Delphine.levilly@univ-grenoble-alpes.fr (D.L.); Denis.wouessi@univ-grenoble-alpes.fr (D.W.); 2Pôle Pharmacie, Grenoble University Hospital, 38000 Grenoble, France

**Keywords:** dexamethasone acetate, cyclodextrins, eye drops, hydrogels, experimental design, phase solubility, dissolution assay

## Abstract

Dexamethasone acetate (DXMa) has proven its efficiency to treat corneal inflammation, without a great propensity to increase intraocular pressure. Unfortunately, its poor aqueous solubility, associated with a rapid precorneal elimination, results in a low drug bioavailability and a low penetration after topical ocular administration. The main objective of this study was to improve the apparent aqueous solubility of DXMa using cyclodextrins. First, hydroxypropyl-β-CD (HPβCD) and hydroxypropyl-γ-CD (HPγCD) were used to enhance DXMa concentration in aqueous solution. The β and γ HPCD derivatives allowed the increase of the DXMa amount in solution at 25 °C by a factor of 500 and 1500, respectively. Second, with the aim of improving the persistence of the complex solution after instillation in the eye, the formulations of DXMa-based CD solutions with marketed ophthalmic gels (CELLUVISC^®^, GEL-LARMES^®^, and VISMED^®^) were investigated and optimized by means of special cubic mixture designs, allowing the defining of mixed gels loaded with 0.7% (HPβCD) and 2% (HPγCD) DXMa with osmolality within acceptable physiological range. Finally, *in vitro* drug release assays from the mixed gels were performed and compared with reference eye drops. Similarly to MAXIDEX^®^ and DEXAFREE^®^, in the case of mixed gel containing HPβCD, more than 90% of the drug was released within 2 h, while in mixed gel containing HPγCD, the release of DXMa was partial, reaching ≈60% in 2 h. This difference will have to be further addressed with *ex vivo* and *in vivo* ocular delivery experiments.

## 1. Introduction

Ocular inflammation is the consequence of many potential eye disorders among which uveitis is believed to be the cause of about 10% of the cases of severe visual handicap in the United States [1]. 

Topical administration of anti-inflammatory drugs, steroidal (SAID) and non steroidal (NSAID), is the most frequently used method to treat ocular surface and anterior segment inflammation as it presents an easy accessibility, a simplicity of use, a non invasive way, and generally a good tolerance. Nevertheless, the ocular drug bioavailability in conventional eye drops is notoriously poor; only 1–5% of drug applied to the surface penetrates the cornea. This is the consequence of various effective protective mechanisms and multiple barriers to drug entry, including a fast naso-lachrymal drainage due to high tear fluid turnover and lid blinking, the corneal structure with a hydrophilic stroma sandwiched between the lipophilic epithelium and endothelium, epithelial drug transport barriers, and clearance from the vasculature in the conjunctiva [2,3].

Numerous strategies have been developed to increase the bioavailability of ophthalmic drugs. One of them is to prolong the contact time between the drug and the corneal/conjunctival epithelium by the use of mucoadhesive hydrogels [4]. An enhanced residence time will increase the time over which absorption can occur and the total amount of drug absorbed and has been shown to result in prolonged effect and increased bioavailability in several studies [5]. As an example, this strategy is used in marketed eye drops such as TIMOPTOL LP^®^ and GELTIM LP^®^, which are instilled once daily *vs.* twice daily with TIMOCOMOD^®^.

Among corticoids, dexamethasone (DXM) has one of the highest potencies and effectiveness on inflammation. DXM is used for the treatment of acute and chronic eye inflammation, including postoperative inflammation or uveitis [6,7]. DXM acts in the human trabecular meshwork cells by inhibiting phospholipase-A2, i.e., prostaglandins synthesis, which causes inflammation [8,9]. Unfortunately, DXM presents a formulation challenge, since it is a water-insoluble compound [10,11]. DXM is soluble to a limited extent in aqueous eye drops. Thus the drug is frequently used as suspensions, such as 0.1% *w*/*v* MAXIDEX^®^ (Novartis Pharma, Rueil-Malmaison, France) or as solutions using a hydrophilic water-soluble prodrug, such as 1% *w*/*v* DXM sodium phosphate DEXAFREE^®^ (Laboratoires Théa, Clermont-Ferrand, France) [7]. The lipophilic derivative DXM acetate (DXMa), currently unavailable for topical use, has been shown to readily permeate the cornea and hydrolyze to DXM during absorption [12]. As well, Leibowitz et al. demonstrated that the acetate form was more effective compared to phosphate derivative in suppressing inflammation in the cornea. This therapeutic effect was not associated with a greater propensity to increase intraocular pressure, one of the most frequent side effects of glucocorticoids [13]. Therefore, DXMa (Figure 1) was selected in this study to be formulated for topical ocular administration. Methods such as pH adjustment, cosolvency, micellization, complexation, or use of cyclodextrins (CD) are among the most commonly used approaches for drug solubilization allowing the formulation of eye drops solution [14]. Cyclodextrins present the great advantage of enhancing both bioavailability and the apparent solubility of poorly water-soluble drugs while being biocompatible [15,16]. In this context, the two hydrophilic cyclodextrin derivatives, hydroxypropyl-β-cyclodextrin (HPβCD) and hydroxypropyl-γ-cyclodextrin (HPγCD), were used to enhance the molecular DXMa fraction in aqueous solution. Mixtures of CD/DXMa solutions with marketed mucoadhesive gels were investigated as topical drug vehicles to the eye, with the objectives to achieve therapeutically effective DXMa dosage form with a reduced frequency of instillation.

In this study, we first evaluated the association constants DXMa/HPβCD or HPγCD. Then, the mixtures of these HPβCD or HPγCD/DXMa solutions with marketed gels (CELLUVISC^®^, GEL-LARMES^®^, and VISMED^®^) were further investigated by means of two mixture experimental designs in order to define optimized DXMa formulations in terms of water-soluble drug fraction content and osmolality. Finally, the *in vitro* release profiles from selected mixed hydrogels were evaluated. 

## 2. Materials and Methods 

### 2.1. Materials

DXMa was purchased from LA COOPER (Melun, France). Hydroxypropyl-γ-cyclodextrin (HPγCD, W8HP, DS = 0.6 and Mw = 1576 Da) was a kind gift from ASHLAND (Schaffhausen, Switzerland) and Hydroxypropyl-β-cyclodextrin (HPβCD, KLEPTOSE DS = 0.63 and Mw = 1391 Da) was obtained from ROQUETTE (Lestrem, France). CELLUVISC^®^ (sodium carboxymethylcellulose), GEL LARMES^®^ (Carbopol 974P) and VISMED^®^ (sodium hyaluronate) are marketed gels used for the treatment of dry eye syndrome. DEXAFREE^®^ (DXM sodium phosphate 1% solution eye drops), MAXIDEX^®^ (DXM 0.1% suspension eye drops) are human authorized ocular medicines. Methanol (HiPerSolvCHROMANORM for HPLC grade) was purchased from BDH, PROLABO (Leuven, Belgium. Purified water was prepared by DIRECT-Q^®^ 3UV water purifier (MILLIPORE, Molsheim, France). All other solvents and chemicals were of HPLC and analytical grade, respectively.

### 2.2. Methods

#### 2.2.1. Quantitative Determinations

Quantitative determinations were performed on a reversed-phase, high-performance liquid chromatographic (HPLC) component system LC 2010 AHT (SHIMADZU, Kyoto, Japan) consisting of a pump with degasser, an autosampler, a UV-VIS detector, and a column XTERRA^©^MS C8 5 µm particles 150 × 4.6 mm with C8 cartridge. This method was adapted from that previously reported by Urban et al. and validated in DXMa, DXM sodium phosphate (DXMp) and dexamethasone (DXM) concentrations ranging from 0.001 to 1 mg/mL [17]. The mobile phase made of methanol:water (70:30 *v*/*v*) was set at the rate of 0.8 mL, the temperature at 25 °C and the detection wavelength at 240 nm. The calibration curves are presented in Table 1.

#### 2.2.2. Phase Solubility Diagrams

The phase solubility studies were carried out according to Higuchi and Connors [18]. Briefly, an excess amount of the drug was added to aqueous solutions containing increasing amounts, 0 to 60% (*w*/*v*) of HPβCD or HPγCD. After 24 h under magnetic stirring at 25 °C, the drug suspensions were ultracentrifugated for 1 h at 35,000 rpm (Optima L-80 XP Ultracentrifuge BECKMAN COULTER, Brea, CA, USA). Note that our operating conditions are limited to 24 h according to preliminary studies, showing that 24 h and 72 h did not change the equilibrium. The supernatant was then diluted at 1:50 in the mobile phase and analyzed by HPLC. The experiments were repeated three times for each cyclodextrin derivative.

The apparent stability constant of the drug/cyclodextrin complex (D/CD), assuming that one molecule of drug forms a complex with one molecule of cyclodextrin (*K*_1:1_), can be calculated from the slope of the linear phase-solubility profiles and the intrinsic drug solubility in the complexation media [18,19] in the absence of the cyclodextrins as presented in Equation (1):
(1)$$K_{1:1}=\frac{\mathrm{Slope}}{S_{0}\times \left(1-\mathrm{Slope}\right)},$$

The complexation efficiency (CE) can be calculated by applying the following Equation (2), which also refers to the slope of the linear phase-solubility profiles [20] and intrinsic solubility:
(2)$$\mathrm{CE}=K_{1:1}\times S_{0}=\frac{\mathrm{Slope}}{\left(1-\mathrm{Slope}\right)},$$

#### 2.2.3. Chromatographic Determination of the Association Constants

The mobile phase consisted of a mixture of methanol:water (70:30 *v*/*v*) with various HPβCD and HPγCD concentrations (0, 0.5, 2.5, 5, 7.5 and 10 mM). Standard solutions of DXMa (0.85 mg/mL) were freshly prepared in a mixture of methanol:water (70:30 *v*/*v*). The chromatographic system was allowed to equilibrate for at least 1 h prior to each experiment. 10 µL of this standard solution was injected and the retention time collected. All the experiments were carried out in triplicate for each temperature (25, 30, 35 and 40 °C) and each cyclodextrin concentration. 

The chromatographic determination of the association constants with high-performance liquid chromatography is based on the partitioning of the solute between the mobile and the stationary phase. When cyclodextrin is added to the mobile phase, solute retention is split into two main physicochemical processes, namely, solute complexation by cyclodextrin and transfer of free (uncomplexed) solute from the mobile to the stationary phase. The association constant *K* (M^−1^) between compound and cyclodextrin can be determined by using the established Equation (3) [21].
(3)$$\frac{1}{k}=\frac{1}{k_{0}}+\frac{K\;{\left[\mathrm{CD}\right]}^{x}}{k_{0}},$$
where *k* (min) is the solute retention factor, *k*_0_ (min) the solute retention factor without cyclodextrin in the mobile phase, [CD] (M) the concentration of cyclodextrin in the mobile phase, and *x* the stoichiometry of the complex. For an inclusion complex with a 1:1 stoichiometry (*x* = 1), a linear plot of 1/*k* versus [CD] must be obtained and the *K* value calculated.

#### 2.2.4. Thermodynamic Parameters for the DXMa/Cyclodextrin Complexes

According to the previous chromatographic conditions, the retention factor was determined in triplicate at the following temperatures: 25, 30, 35 and 40 °C. *ΔH°* and *ΔS°* are, respectively, the standard enthalpy and entropy of transfer of DXMa from the mobile phase to the cyclodextrin cavity. These energies can be calculated using the following thermodynamic relationships as described in Equation (4) [21,22]: (4)$$\ln K=\frac{-\Delta H{}^{\circ}}{RT}+\frac{\Delta S{}^{\circ}}{R},$$
where *T* is the temperature and *R* the gas constant. For a linear plot of ln*K* versus 1/*T*, the slope and the intercept are, respectively −Δ*H*°/*R* and Δ*S*°/*R*.

#### 2.2.5. Experimental Designs and Data Analysis

Experimental designs were used in order to determine the optimized formulations based on HPβCD and HPγCD. In this study, the goal of optimization was respectively focused on maximization of the DXMa solubility and adjustment of osmolality within acceptable physiological range from 250 to 450 mOsm/Kg [23]. Experimental domain was obtained by fixing the minimum and maximum proportions of each component (% *w*/*w*) as presented in Table 2. 

Design-Expert software version 10.0.8 (State Ease, Inc., Minneapolis, MN, USA) package was used to establish a special cubic mathematical model which exhibits the relationships between response and formulation components, allowing the optimum operational conditions to be obtained via a statistical analysis.

Two experimental designs (1 and 2) were performed, one per CD derivative (Table 2). Each experimental design included 29 experiments. Each experiment was performed according to the procedure described in Figure 2. Briefly, a mixture (2 g) containing DXMa/CD solution and hydrogel(s) was stirred for 2 h under magnetic stirring at room temperature. Then, a large excess of DXMa was added to the mixture and agitated during 12 h. The drug suspension was ultracentrifugated at 15 °C during 1h at 35,000 rpm (Optima L-80 XP Ultracentrifuge BECKMAN COULTER, Brea, CA, USA). The supernatant was collected and the osmolality measured (Model 2020, ADVANCED INSTRUMENT, Norwood, MA, USA) before a dilution at 1:50 with the mobile phase in order to assay the rate of DXMa by HPLC.

The special cubic model coefficients were estimated in accordance with the established Multi Linear Regression (MLR), which allows fitting of the observed response with the analytical model [24]. 

The full mixture cubic model including all coefficients was refined using stepwise technique [25]. This procedure involves removing step by step each eligible coefficient to find the model that best fits the data according to some criteria. The corrected Akaike Information Criterion (AICc) and the Bayesian Information Criterion (BIC) minimization are the likelihood statistics criteria used to compare the different models.

The fitness of the models can be validated using statistical parameters as R-square (*R*^2^), adjusted R-square ($R_{adj}^{2}$), predicted R-square ($R_{pred}^{2}$), and adequation precision (*AdeqPrec*) values. The *R*² value, as shown in Equation (5), refers to the ratio of the sum of squares regression (*SS_R_*) to the total sum of squares (*SS_T_*) from the ANOVA table. The *R*² value explains the total variation of the data around the average, and its value is in the range of 0–1.0. A value of *R*² close to 1.0 indicates that the models have good fit. Nonetheless, the value of *R*² is directly related to the number of terms in the model. Therefore, the additional checking criteria ($R_{adj}^{2}$) and ($R_{pred}^{2}$) are also needed (Equations (6) and (7)). In Equation (6), *p* denotes the number of factors plus one and SS_E_ is the error or residual sum of squares while PRESS in Equation (7) is the predicted residual error sum of squares. Generally, $R_{adj}^{2}$ decreases as insignificant terms are added to the model and $R_{pred}^{2}$ decreases when the model considers too many insignificant terms. Therefore, these two criteria are the primary concerns in response modeling, where both values should be close to 1.0 and within 0.2 of one another [26]. The adequation precision (*AdeqPrec)* measures the signal-to-noise ratio (Equation (8)). ${\hat{\sigma }}^{2}$ denotes the residual mean square from ANOVA table, $\max\;\left(\hat{Y}\right)\mathrm{and}\;\min\left(\hat{Y}\right)$, respectively, are the maximal and minimal response predicted for the experimental design run conditions. A ratio greater than 4 is desirable.
(5)$$R^{2}=\frac{SS_{R}}{SS_{T}},$$
(6)$$R_{Adj}^{2}=1-\frac{\frac{SS_{E}}{n-p}}{SS_{T}\left(n-1\right)}=1-\frac{n-1}{n-p}\left(1-R^{2}\right),$$
(7)$$R_{pred}^{2}=1-\frac{PRESS}{SS_{T}},$$
(8)$$AdeqPrec=\frac{\max\;\left(\hat{Y}\right)-\min\left(\hat{Y}\right)}{\sqrt{\frac{p{\hat{\sigma }}^{2}}{n}}},$$

In complement, scatter plots of Actual *vs.* Predicted are used to evaluate how the model predicts over the range of data. Ideally, the predicted values should be close to the actual values and then all points should be close to a regressed diagonal line. Furthermore, the points should be symmetrically scattered about the line, as expected if the errors are normally distributed.

#### 2.2.6. Rheological Characterization

The viscosity of optimized mixed gels A and B (*n* = 3) were determined by using a rotational viscometer (RM 100, LAMY, Champagne au Mont d’Or, France). The viscosity measurements were performed at controlled temperature (22 °C) at increasing shear rates (from 12.9 to 1936 s^−1^).

#### 2.2.7. *In Vitro* DXMa Release Profiles

The drug release experiments were carried out using a Sotax Dissolutest AT7 (SOTAX, Aesch, Switzerland). A sample of optimized mixed gels A or B or MAXIDEX^®^ or DEXAFREE^®^ was dropped in the extraction cell, which was placed at the bottom of the vessel filled with the dissolution medium. The experiments were conducted for 24 h at 35 °C, in 250 or 500 mL of phosphate buffer saline (PBS 1X pH 7.4). The speed of the rotating paddle was set at 100 rpm. The DXM, DXMa, and DXMp solubilities were previously determined in triplicate after 2 h agitation of aqueous drug suspensions in PBS at 35 °C. After filtration (0.2 µm), the solubilized drug content was quantified by HPLC at 240 nm. The amounts of sample used in the cell were 1.5 g for gel A, MAXIDEX^®^, and DEXAFREE^®^, and 0.5 g for gel B. At the set time points (30 min, 1 h, 2 h, 4 h, 8 h, 24 h), aliquots of 1mL filtered medium were withdrawn and DXMa content measured by HPLC.

## 3. Results and Discussion

DXMa, a poorly water-insoluble steroid, was selected in this study to be formulated for topical ocular administration. In this context, the two hydrophilic cyclodextrin derivatives, HPβCD and HPγCD, were used to enhance the molecular DXMa fraction in aqueous solution.

### 3.1. Solubility Determinations of Dexamethasone Acetate

The phase-solubility study is one of the most common methods applied in order to evaluate the solubilization ability of CDs. Figure 3 was obtained by plotting the total concentration of dissolved cyclodextrin (mM) HPβCD or HPγCD versus apparent DXMa concentrations at equilibrium (mM). The obtained profiles were then classified according to Higuchi and Connors [18]. For both cyclodextrin derivatives, the phase-solubility profiles are linear (*R*^2^ ≥ 0.995), indicating that the apparent solubility of DXMa increases with an increase of the cyclodextrin concentration. Thus these linear curves refer to AL-type phase-solubility profiles according to Higuchi and Connors [18]. The solubility studies indicated that the DXMa probably forms water-soluble complexes with the two CDs. Indeed, our results showed a dramatic increase of DMXa solubility induced by the complexes. Typically, 600 mg/mL (380 mM) HPβCD and HPγCD (430 mM) aqueous solutions at 25 °C solubilize 10.91 ± 0.16 mg/mL and 30.48 ± 0.12 mg/mL of DXMa, respectively, which correspond to increasing in solubility of about 520- and 1450-fold compared to aqueous solubility of uncomplexed DXMa at 25 °C (i.e. *S*_0_ = 0.021 mg/mL). Usayapant et al. and Vianna et al. also studied interaction between DXMa and cyclodextrins [12,27]. Especially, Usayapant et al. found that at 260 mM HPβCD, the solubility enhancement for DXMa was 1016-fold. For their part, Vianna et al. indicated a 88-fold increase of DXMa with a maximum of 53 mM HPβCD. The differences between these solubility values are probably related to the experimental conditions of phase-solubility studies such as pH, ionic strength, temperature, the time necessary to reach equilibrium, the range of CD concentration, the graphical determination of *S*_0_ and analytical method used. The degree of substitution of the HPβCD used is also to be considered. Despite these differences, HPβCD complexation allowed the DXMa water solubility to be enhanced significantly. So far there are few studies concerning complexation of HPγCD with DXMa, hence our study, which showed a remarkable increase of apparent solubility of DXMa up to 30.48 mg/mL in the presence of 600 mg/mL HPγCD, is of promising interest. Concerning the type of phase-solubility diagram, according to our solubility profiles, we assumed that both cyclodextrin derivatives lead to A_L_-type. It is to highlight that Vianna et al. also described an A_L_-type phase-solubility profile for DXMa/HPβCD complex while Usapayant et al. claimed an A_P_-type phase-solubility profile.

The stoichiometric and binding or association constant *K*, as well as the complexation efficiency (CE), are important characteristics of the complex. Based on the phase A_L_-type phase-solubility diagram, a stoichiometry of 1:1 was assumed for DXMa/HPβCD and DXMa/HPγCD. Usayapant et al. claimed that 1:1 and 1:2 complexes where present when HPβCD interacted with DXMa in solution [12]. However, Usayapant et al. also indicated in their work that the formation of 1:2 complex was less favored due to the very low value of the constant binding *K*_1:2_ inferior to 20 M^−1^. When referring to the literature, some authors have described the geometry of inclusion complexes of steroids with cyclodextrins. Usually it is reported that inclusion occurred primarily at the A–B ring, especially when A ring bears a ketone (Figure 1) [27,28,29]. Concerning specially the DXMa, it is not totally excluded that the acetyl group can interact with the cyclodextrin cavity [12,27].

In this study, assuming the formation of 1:1 complex, the apparent stability constant of the DXMa/cyclodextrin complex was first calculated from the slope of the linear phase-solubility profiles and the intrinsic drug solubility (*S*_0_ = 0.021 mg/mL = 0.048 mM) in the complexation media [18] in the absence of the cyclodextrins, as presented in Equation (1).

The complexation efficiency (CE) was calculated by applying Equation (2), which also refers to the slope of the linear phase-solubility profiles [20] and intrinsic solubility of DXMa.

The results of calculated *K*_1:1_ and CE, as well as the slope of phase-solubility diagrams, are shown in Table 3. The *K*_1:1_ values were 1462 and 5368 M^−1^ for DXMa/HPβCD and DXMa/HPγCD, respectively. The *K*_1:1_ value reported by Usayapant et al. for DXMa/HPβCD was 2240 M^−1^ and slightly higher. This difference can be explained by the fact that phase-solubility studies are influenced by various factors such as the operating conditions, that is, pH, ionic strength, temperature, and analytical method, that do not allow a strict comparison of binding constants *K* obtained from different studies. The *K* value of DXMa/HPγCD complex is about four times higher than that of DXMa/HPβCD complex. This result could be explained by the larger size of the HPγCD 8.3 Å, which better accommodates with the A–B ring, while the size of HPβCD cavity is smaller, 6.5 Å, to allow a strong interaction with this cyclodextrin [29]. So far, *K* value of DXMa/HPγCD complex has not been reported in the literature, however, a comparison could be done at least with the binding constant of the complex DXM/HPγCD equal to 5190 M^−1^, determined by Jansook et al. 

Regarding the pharmaceutical applications of cyclodextrins, it is important to choose the derivative exhibiting the higher solubilizing efficiency. The complexation efficiency was calculated as 0.071 and 0.259 for DXMa/HPβCD and DXMa/HPγCD complexes, respectively. CE of 0.071, approximately 0.1, suggests that 1 out of 11 HPβCD molecules forms a complex with DXMa, and CE of 0.259, approximately 0.3, suggests that 3 out of 4 HPγCD molecules are involved in forming a complex with DXMa [30]. From a strict point of view of the drug formulation, it would therefore be advantageous to choose HPγCD as host agent instead of HPβCD.

### 3.2. Chromatographic Determination of the Association Constants Between Dexamethasone Acetate and HPβCD or HPγCD

Using the solute retention time and the void time, the *K* values were determined for all the cyclodextrin concentrations at temperatures of 25, 30, 35 and 40 °C. The coefficients of variation of the *k* values were <0.5%, indicating a high reproducibility and a good stability for the chromatic system. The 1/*k vs.* [HPβCD] or [HPγCD] plots were determined and the values of the linear regression coefficients *R²* were calculated. The *R*² values were higher than 0.934 in all cases. For example, Figure 4 shows the two plots corresponding to the two cyclodextrin derivatives at 40°C. The results of the association constants *K* are presented in Table 4 together with those found in the literature. From these results, it appears clearly that the interaction of DXMa with the two cyclodextrin derivatives is well described by the 1:1 stoichiometry model as claimed by other authors [12,27]. As expected, our results showed that *K* values of DXMA/HPγCD complex are higher than those of DXMa/HPβCD complex. We find that the *K* value of DXMa/HPγCD obtained from phase-solubility diagram at 25 °C is about two fold higher than that calculated by chromatographic study. Concerning the complex DXMa/HPβCD, our *K* value obtained from phase-solubility studies is not very far from that reported by Usayapant et al. [12].

Finally and regardless of the characterization methods implemented, it is clear that the affinity of DXMa for HPγCD is greater than that for HPβCD. 

### 3.3. Thermodynamic Parameters for the DXMa/Cyclodextrin Complexes

In order to gain information about the mechanistic aspect of the difference in the solute affinity for HPβCD and HPγCD, the thermodynamic parameters were obtained from Van’t Hoff plots. The ln*K* vs. 1/*T* plots were obtained for the two cyclodextrins. Figure 5 shows linear Van’t Hoff plots with correlation coefficient higher than 0.988. Table 5 presents Δ*H*° and Δ*S*° for the two complexes with the corresponding Gibbs free energy Δ*G*° at 25 °C.

For both DXMa/HPβCD and DXMa/HPγCD associations, Δ*H*° exhibits weak negative values while Δ*S*° ones are positive. These values demonstrate that the association phenomenon is both enthalpically and entropically driven (Table 5). At 25 °C, in the case of DXMa/HPβCD, the contributions of enthalpic and entropic terms to the Gibbs free energy are almost identical, suggesting that the DXMa/HPβCD association is dependent on the hydrophobic effect between non polar groups of solute and the hydroxypropyl groups of the cyclodextrin derivative. Similar observations have been reported with NSAID association and HPβCD [21]. In the case of DXMa/HPγCD, the contribution of the enthalpic term to the Gibbs free energy is higher, close to 70%. 

All the results described above clearly indicated that complexation between HPβCD or HPγCD with DXMa significantly increases the water solubility of the guest molecules. Although the complexation efficiency of HPβCD was lower than that of HPγCD, we decided to continue the eye drop formulation studies keeping the two pairs of complexes DXMa/HPβCD and DXMa/HPγCD. It should be noted that the European Pharmacopoeia and USP/NF have published monographs for HPβCD. The natural γCD has a monograph in the Japanese Pharmaceutical Codex and the USP/NF. So far, HPγCD has not been registered as excipient by any of the major Pharmacopoiea [15]. Nevertheless, one can note that this derivative is already used in the composition of the marketed eye drop VOLTAREN OPHTHA^®^. 

### 3.4. Special Cubic Mixture Designs

Using the DXMa solubility and osmolality of each experiment (Appendix A), the experimental designs for both cyclodextrin derivatives were analyzed using Design-expert software and the results are reported in Table 6.

Based on these results, the *R*^2^, $R_{adj}^{2}$, and $R_{pred}^{2}$ values for the osmolality and [DXMa] for HPβCD are higher than 0.98, indicating that the refined model has high regression accuracy. For HPγCD, the *R*^2^,
$R_{adj}^{2}$, and $R_{pred}^{2}$ statistics values are higher than 0.85, indicating that the refined model has good regression accuracy. For all considered responses, the *AdeqPrec* values are up to 31.45, which indicates an adequate signal to noise. The analysis of variance (ANOVA) results are described in Appendix A and evidenced that the reduced models were highly significant (*p*-value < 0.05) [31]. The scatter plots of Actual *vs*. Predicted responses are useful to detect misspecifications in the structural model. Here, this figure does not reveal any significant bias. We can observe that the points are lying around the line along the total length of the line, that the amount of variation around the line does not change along the length of the line, and that there are no outliers.

In addition to the experimental design points, a set of supplementary trials at a single combination of factors settings are added to ensure the accuracy of the reduced mixtures models. The desirability function proposed by Derringer and Suich [32] is used to realize the simultaneous optimization of both osmolality and DXM solubility responses. In this study, the goal of optimization was respectively focused on maximization of the DXM solubility and adjustment of osmolality within acceptable physiological range. A Nelder–Mead simplex-algorithm-based numerical optimization is used to identify the best subset of variable setting combination that maximizes the desirability function [33,34]. Finally, the selected levels of variables used as the model confirmation samples are reported in Table 7.

The average of response (*n* = 2) of the confirmation sample is compared to the 95% prediction interval. For both cyclodextrin derivatives, the reduced cubic models for osmolality and [DXMa] are experimentally validated because the average observation of the supplementary experiments proposed in the Table 7 are within the confirmation node’s prediction interval.

Table 8 reports the quantitative composition of two mixed gels containing either HPβCD or HPγCD, resulting from the experimental designs, achieving high DXMa content and acceptable osmolality. These formulations are further denoted optimized mixed gels A (HPβCD) and B (HPγCD).

### 3.5. Rheological Characterization

The administration of an ophthalmic formulation should not influence the pseudoplastic nature of precorneal film, or the influence should be negligible. Figure 6 shows the apparent viscosity of the mixed gels A and B as a function of shear rate. They both showed pseudo plastic behavior. The apparent viscosity value was lower as the speed gradient increased. At 22 °C and 12.9 s^−1^ (the lowest shear rate allowing stable value to be obtained), the mean ± SD apparent viscosity (*n* = 3) of optimized formulation gels A and B was 98 ± 2 mPa·s and 78 ± 1 mPa·s, respectively. 

This viscosity range and the non-Newtonian behavior is fruitful for ophthalmic use due to the fact that the ocular shear rate is highly variable, ranging from 0.03 s^−1^ during interblinking periods to 4250–28,500 s^−1^ during blinking. If the viscosity at a high shear rate is too high, this will result in irritation. On the other hand, if the viscosity is too low, it will give rise to increased drainage. The rheological property of these formulations should be in favor of sustaining drainage of drugs from the conjunctival sac of the eye without blinking difficulty in undergoing shear thinning [6,35].

### 3.6. In Vitro DXM Release Studies

*In vitro* release assessment was performed on several formulations, namely, both optimized mixed gels A and B, as well as two reference marketed eye drops, MAXIDEX^®^ and DEXAFREE^®^. The drug solubilities were, respectively, 0.1 mg/mL, 0.015 mg/mL, and higher than 10 mg/mL for DXM, DXMa, and DXMp in PBS at 35 °C. On this basis, and in order to ensure sink condition dissolution testing, all the experiments were performed in sufficient volume media related to either DXM, DXMa, or DXMp to be dissolved. Therefore, the highest concentration corresponding to 100% drug release could not exceed 0.05 mg/mL, 0.01 mg/mL, and 5 mg/mL for DXM, DXMa, and DXMp, respectively, so that the dissolution media solution would not reach saturation. As shown in Figure 7, a complete drug release was observed at 24 h (1440 min) for DEXAFREE^®^ with the major part of drug released within 30 min (92%). As well, MAXIDEX^®^ exhibited similar DXM release profile with 90% of the drug recovered in the release medium at 30 min and 10% remaining released over 24 h. Optimized mixed gel A exhibited DXMa release comparable to the reference eye drops, the mixed gel A being superimposable to the MAXIDEX^®^ one. The release experiments observed for mixed gel A showed that upon high dilution conditions, the DXMa molecules could freely diffuse from the delivery system, which is a prerequisite for local biological activity. When looking at the mixed gel B behavior, 56% of the drug diffused in the external medium after 2 h. The missing DXMa fraction was recovered in the cell extraction, meaning that a part of the mixed gel B remained stuck to the cell surface during the experiment, limiting further DXMa release. Indeed, the amount of DXMa very slightly increased with time, achieving 60% and 62% after 24 h and 48 h, respectively (unshown result). This retention phenomenon was only observed in the case of HPγCD containing mixed gel. This characteristic will have to be addressed in future *in vivo* bioremanence studies.

## 4. Conclusions and Future Prospects

The formulation of dexamethasone acetate, a highly lipophilic corticosteroid prodrug, was investigated for topical ocular delivery. High drug contents in aqueous solution were achieved by using HPβCD or HPγCD at a concentration of 600 mg/mL, allowing the increase by a factor of around 500 and 1500, respectively, of the DXMa amount in water at 25 °C. The mixtures of these HPβCD or HPγCD/DXMa solutions with marketed gels were further investigated by means of two mixture experimental designs. New mixed gels loaded with 0.7% and 2% DXMa were developed made of sodium hyaluronate and/or carbopol with HPβCD or HPγCD, respectively. Both mixed gels released the drug *in vitro* after dilution in PBS at 35 °C, more or less completely depending on the composition of the vehicle. Next steps of the study will focus on mucoadhesion properties of DXMa mixed gel formulations as well as cytotoxicity studies. The statistical approach by experimental designs and the good prediction power of the models will be helpful to further adjust the compositions of the mixed gels as a function of future *in vitro* and *in vivo* results.

## Figures and Tables

**Figure 1 pharmaceutics-10-00249-f001:**
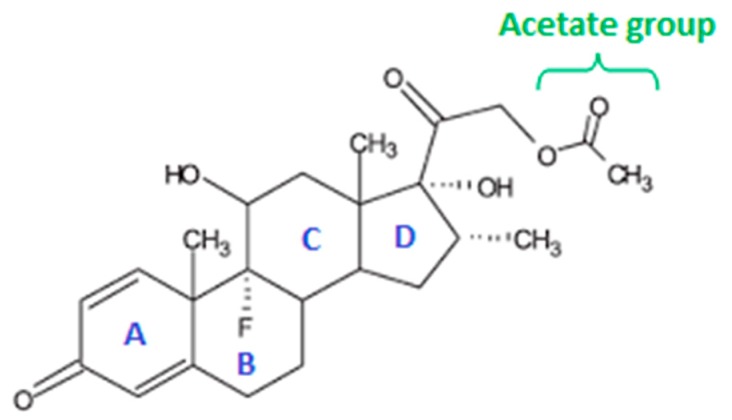
Chemical structure of dexamethasone-21-acetate with perhydro-cyclopentano-phenanthrene ring system, A = A-ring, B= B-ring, C= C-ring, and D = D-ring.

**Figure 2 pharmaceutics-10-00249-f002:**
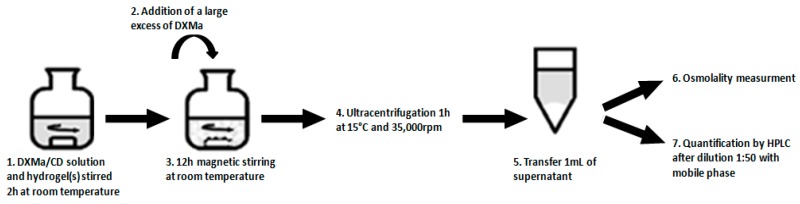
Steps implemented for the experimental design with 29 experiments.

**Figure 3 pharmaceutics-10-00249-f003:**
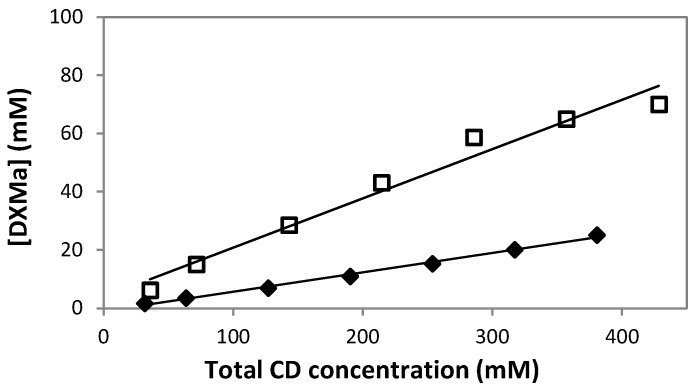
Phase-solubility diagrams of DXMa in water under various concentrations of HPβCD (
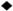
) or HPγCD (
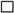
). Each data point represents a mean (*n* = 3), with SD smaller than the symbol size.

**Figure 4 pharmaceutics-10-00249-f004:**
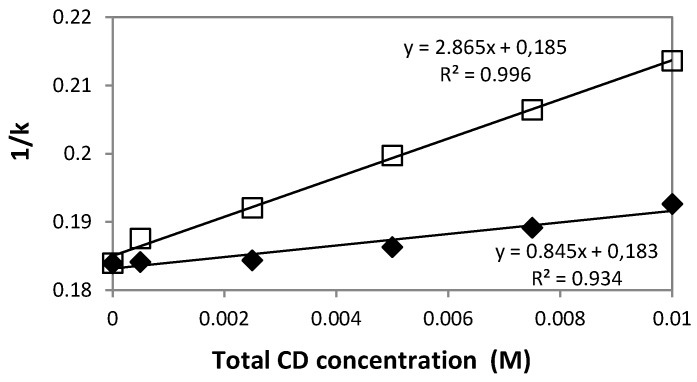
Plots of 1/*k vs.* [HPβCD] (
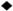
) or [HPγCD] (
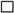
) (assuming 1:1 stoichiometry) for dexamethasone acetate at a column temperature equal to 40 °C. Stationary phase: phenyl silica gel; mobile phase: mixture methanol: water (70:30 *v*/*v*).

**Figure 5 pharmaceutics-10-00249-f005:**
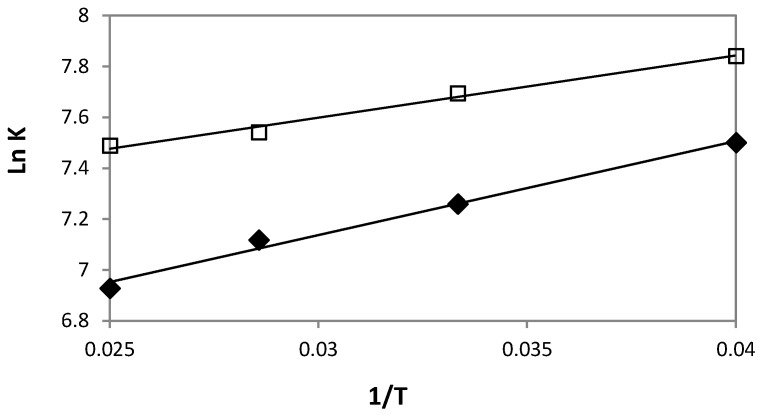
Van’t Hoff plots (ln*K* vs. 1/*T*) for DXMa/HPβCD (
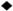
) or DXMa/HPγCD (
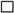
) associations.

**Figure 6 pharmaceutics-10-00249-f006:**
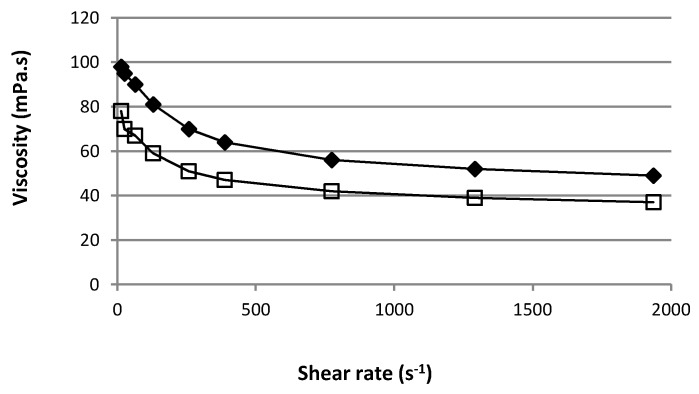
Rheological profiles of the two optimized mixed gels based on HPβCD (
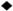
) or HPγCD (
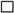
).

**Figure 7 pharmaceutics-10-00249-f007:**
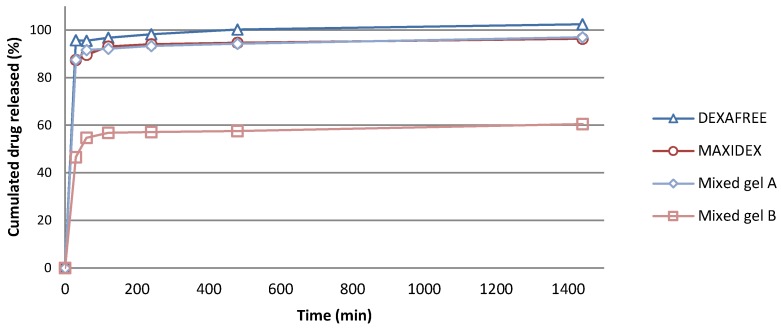
*In vitro* drug release from MAXIDEX^®^, DEXAFREE^®^ and optimized mixed gels A and B in PBS, at 35 °C.

**Table 1 pharmaceutics-10-00249-t001:** Calibrations curve, retention time, correlation coefficient and variability of DXM, DXM sodium phosphate and DXMa quantitative determinations by HPLC.

Drugs	Retention Time	Calibration Curve	Correlation Coefficient	Intra-Day Variability (CV%)	Inter-Day Variability (CV%)
DXM	4.8	y = (3 × 10^7^)x + 27.867	0.999	<1%	<3%
DXM sodium phosphate	3.8	y = (5 × 10^6^)x – 312.7	0.999	<1%	<2%
DXMa	6.3	y = (3 × 10^7^)x + 39.464	0.999	<1%	<3%

**Table 2 pharmaceutics-10-00249-t002:** Low and high levels of formulation components for special cubic mixture designs.

Component		Low Level (%)	High Level (%)	
Experimental Design 1	CELLUVISC^®^-Gel_1_	0	70	
	GEL-LARMES^®^-Gel_2_	0	70
	VISMED^®^-Gel_3_	0	70
	HPβCD 600 mg/mL with DXMa 10 mg/mL	30	100
Experimental Design 2	CELLUVISC^®^-Gel_1_	0	70
	GEL-LARMES^®^-Gel_2_	0	70
	VISMED^®^-Gel_3_	0	70
	HPγCD 600 mg/mL with DXMa 30 mg/mL	30	100

**Table 3 pharmaceutics-10-00249-t003:** Apparent stability constant *K* and the complexation efficiency (CE) of DXMa/cyclodextrin complexes at 25 °C.

CD Type	Slope	Correlation Coefficient	*K*_1:1_ (M^−1^)	CE
HPβCD	0.066	0.995	1462	0.071
HPγCD	0.206	0.999	5368	0.259

**Table 4 pharmaceutics-10-00249-t004:** Apparent association constants *K* of the complexes DXMa/HPβCD and DXMa/HPγCD determined by chromatographic procedure at various temperatures compared to literature data.

Method	Chromatographic Experiments				Phase Solubility Studies		UV Spectroscopy
**Reference**	**Present study**				**Present study**	[12]	[12]
**Solution**	**methanol:water (70:30)**				**water**	**water 0.1 M citrate buffer (pH 6.0)**	**water 0.1 M citrate buffer (pH 6.0)**
**Temperature (°C)**	**25**	**30**	**35**	**40**	**25**	**25**	**25**
HPβCD	1807	1421	1234	1020	1462	2240	2445
HPγCD	2541	2195	1883	1787	5368	-	-

**Table 5 pharmaceutics-10-00249-t005:** Thermodynamic parameters Δ*H*°, Δ*S*° and Δ*G*° at 25 °C for DXMa/HPβCD and DXMa/HPγCD complexes.

DXMa/CD Complexes	Δ*H*°		Δ*S*°		Δ*G*° (kJ/mol)
	kJ/mol	Contribution to Δ*G*°	J/mol K	Contribution to Δ*G*°	
DXMa/HPβCD	−20.3	54%	+57.1	46%	−3.3
DXMa/HPγCD	−30.7	67%	+50.1	33%	−15.7

**Table 6 pharmaceutics-10-00249-t006:** Best models containing the best subset of the predictors after backward stepwise selection, overall quality of model fit and the corresponding predicted against actual plot, Gel_1_: CELLUVISC^®^, Gel_2_: GEL-LARMES^®^ and Gel_3_: VISMED^®^.

		Final Equation in Terms of Actual Components	Model Evaluation	Predicted vs. Actual Plot
**HPβCD**	**Osmolality**	Osmolality(mOsm/Kg) = +342.68 × Gel_1_ +369.84 × Gel_2_ +242.96 × Gel_3_ +765.94 × [HPβCD] −504.45 × Gel_1_ × [HPβCD] −452.83 × Gel_2_ × [HPβCD] −757.14 × Gel_3_ × [HPβCD]	*R*^2^ = 0.9900*R^2^_adj_* = 0.9872 *R^2^_pred_* = 0.9849 *Adeq Prec* = 98.87 BIC = 232.56 AICc = 228.17	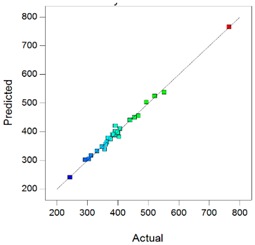
	**[DXMa]**	[DXMa] (mg/mL) = (+1.47×10^−3^) × Gel_1_ +0.50 × Gel_2_ +1.16 × Gel_3_ +10.90 × [HPβCD] −2.93 × Gel_1_ × [HPβCD] −6.01 × Gel_2_ × Gel_3_ −4.29 × Gel_2_ × [HPβCD] −5.49 × Gel_3_ × [HPβCD] +17.45 × Gel_2_ × Gel_3_ × [HPβCD]	*R*^2^ = 0.9980 *R^2^_adj_* = 0.9972 *R^2^_pred_* = 0.9932 *Adeq Prec* = 155.01 BIC = −37.76 AICc = −41.50	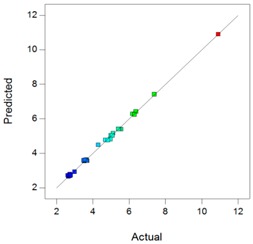
**HPγCD**	**Osmolality**	Osmolality (mOsm/Kg) = +768.79 × Gel_1_ +57.80 × Gel_2_ +329.10 × Gel_3_ +789.73 × [HPγCD] −1367.82 × Gel_1_ × Gel_2_ −1986.36 × Gel_1_ × [HPγCD] −140.20 × Gel_2_ × [HPγCD] −1053.47 × Gel_3_ × [HPγCD] +5990.04 × Gel_1_ × Gel_2_ × [HPγCD]	*R*^2^ = 0.9403 *R^2^_adj_* = 0.9165 *R^2^_pred_* = 0.8517 *AdeqPrec* = 31.45 BIC = 297.68 AICc = 293.94	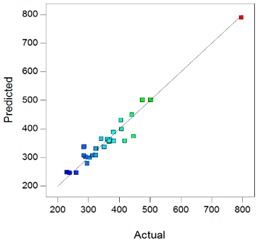
	**[DXMa]**	[DXMa] (mg/mL) = +4.36 × Gel_1_ −5.70 × Gel_2_ −4.60 × Gel_3_ +29.10 × [HPγCD] −20.34 × Gel_1_ × [HPγCD]	*R*^2^ = 0.9450 *R^2^_adj_*= 0.9357 *R^2^_pred_* = 0.9055 *AdeqPrec* = 42.94 BIC = 108.39 AICc = 104.58	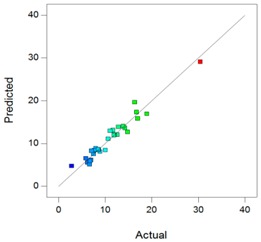

**Table 7 pharmaceutics-10-00249-t007:** Variables setting combination used for models confirmation samples (*n* = 2).

CD Type	Gel_1_ CELLUVISC^®^ (%)	Gel_2_ GEL-LARMES^®^ (%)	Gel_3_ VISMED^®^ (%)	CD (%)	Actual Osmolality	Predicted Osmolality	Actual[DXMa] (mg/mL)	Predicted[DXMa] (mg/mL)
HPβCD	0.000	0.000	0.300	0.700	429	450.045	6.973	6.826
HPβCD	0.000	0.145	0.215	0.640	450	449.858	6.319	6.305
HPβCD	0.454	0.000	0.000	0.546	435	448.735	4.651	5.226
HPγCD	0.089	0.089	0.098	0.724	519	489.326	17.153	19.188
HPγCD	0.000	0.425	0.000	0.575	447	444.396	12.813	14.310
HPγCD	0.244	0.201	0.000	0.555	436	448.831	13.492	13.314

**Table 8 pharmaceutics-10-00249-t008:** Composition of optimized mixed gels A and B.

	Components	Quantity (g)
**Optimized mixed gel A**	VISMED^®^-Gel_3_	0.300
	HPβCD 600 mg/mL with DXMa	0.700
	Optimized mixed Gel A contains 7 mg/g of DXMa and an osmolality of 449 mOsm/kg	
**Optimized mixed gel B**	CELLUVISC^®^-Gel_1_	0.151
	VISMED^®^-Gel_3_	0.085
	HPγCD 600 mg/mL with DXMa	0.764
	Optimized mixed gel B contains 20 mg/g of DXMa and an osmolality of 425 mOsm/kg	

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
