# Peer review of "Investigation of Combined Cyclodextrin and Hydrogel Formulation for Ocular Delivery of Dexamethasone Acetate by Means of Experimental Designs"

_pharmaceutics, 2018, doi:10.3390/pharmaceutics10040249_

Reviewer 1 Report

This is a very good paper about the preparation of eye solutions of Dexamethasone acetate for use in the treatment of corneal inflammation. The manuscript is well written and the work well developed and organized and I consider that has the quality and interest to be published in Pharmaceutics.

One suggestion for the authors is include some information about the safety of the use of HPγCD in the eye.

Author Response

Concerning the safety of use of HPγCD, this derivative is present as excipient in the composition of marketed eye drug dosage form containing diclofenac.The sentence was added in the revised version line 360 : Nevertheless, one can note that this derivative is already used in the composition of the marketed eye drop VOLTAREN OPHTHA®.

Reviewer 2 Report

Cyclodextrin-dextamethasone acetate complexation was first investigated to improve the solubility of the drug.  A special cubic mixture design was then used to optimize the gel formulation containing the drug cyclodextrin complexes.  The developed formulations were finally compared with two commercial products for drug release.  In addition to HP-beta-CD, this research also studied the application of HP-gamma-CD in ocular delivery.  Overall, the manuscript may be shortened and written more concisely.  Some data and figures may be included as supplemental materials or removed.  There are typos and grammatical errors. 

1.       Section 2.2.7 is not clear.  For example, “The amount of sample (0.5 to 1.5 g) used in the cell was adjusted in each case as a function of drug solubility in the external medium”.  Experimental details should be added. 

2.       The release profiles in page 15 are similar except mixed gel B.  What are the advantages of the developed formulations compared to the commercial products?

3.       Page 15, drug release profiles should be Figure 8 instead of Figure 2

4.       Figure 3 is not necessary and thus can be removed.

5.       Standard deviations are missing for some data, for example, 30.48 mg/mL (Page 7, line 214)

Author Response

1-      Section 2.2.7 is not clear.  For example, “The amount of sample (0.5 to 1.5 g) used in the cell was adjusted in each case as a function of drug solubility in the external medium”.  Experimental details should be added.

The sentence line 222 was modified as follows: The amounts of samples used in the extraction cells were 1.5 g for gel A, MAXIDEX® and DEXAFREE® and 0.5g for gel B.

2-      The release profiles in page 15 are similar except mixed gel B.  What are the advantages of the developed formulations compared to the commercial products?
The solubilized drug concentrations in the gels A (0.7%) and B (2%) are higher than commercial products, increasing the available drug fraction, which is proved to be in vitro released from gels A (95%) and B (60%) in 2 hours. The amount of drug corresponding to 60% released from gel B is higher than that of 100% MAXIDEX® or DEXAFREE®.

3-      Page 15, drug release profiles should be Figure 8 instead of Figure 2

The number of the figure has been modified.

4-      Figure 3 is not necessary and thus can be removed. 
As suggested by the reviewer, the figure 3 is removed.

5-      Standard deviations are missing for some data, for example, 30.48 mg/mL (Page 7, line 214)

The mean ± SD “30.48 ± 0.12 mg/mL” is added in the revised version (line 242)

Reviewer 3 Report

1.   The abstract appears incomplete. No mention of in vitro results, or concluding statements could be found.

2.   The condition of “ocular inflammation” is not well defined in the introduction. The authors are asked to elaborate on specific types of inflammation e.g. conjunctivitis, uveitis, etc. and add some statistics or references to justify their claims of the disorder being “common”.

3.   Lines 38-40: This paragraph needs to be larger with more elaboration about types of hydrogels and what has been found with using these so far, rather than jumping straight into the ‘corticoids’ paragraph.

4.   Line 53: This sentence does not do a good job justifying the use of DXMa in the study.

5.   Figure 1: the labelling of “rings” looks messy. The authors are requested to revise the style of arrows used.

6.   Reference 14 has been unnecessarily placed within the methods. Reference 23 in the results and discussion section also appears to be unnecessarily placed.

7.   Please substitute “well-known” with “established”.

8.   Figure 2: Specify within the figure what is in the beaker/container during step 1 (i.e. the complex).

9.   Line 161: Why is Table 6 mentioned before tables 3-5?

10.Equations are interspersed between both the ‘methods’ and ‘results and discussion’ sections. The authors are advised to keep ALL equations within the ‘methods’ section.

11.Table 9 and Figure 7: is the DXMa content consistent in all the gels tested? Can gels with unequal drug concentrations be compared?

Author Response

 1-       The abstract appears incomplete – no mention of in vitro results or concluded statement could be found.

 As suggested, the abstract has been modified. Line 22, the following sentence has been added : Similarly to MAXIDEX® and DEXAFREE®, in the case of mixed gel containing HPβCD, more than 90% of the drug was released within 2 hours while in mixed gel containing HPγCD, the release of DXMa was partial reaching ≈ 60% in 2 hours. This difference will have to be further adressed with in vivo experiments.

 2-       The condition of “ocular inflammation” is not well defined in the introduction. The authors are asked to elaborate on specific types of inflammation e.g. conjunctivitis, uveitis, etc. and add some statistics or references to justify their claims of the disorder being “common”.

 As suggested, a reference is added line 33. As well, the sentence « DXM is used for the treatment of acute and chronic eye inflammation, including post-operative inflammation or uveitis [6,7] » is added line 49.

3-       Lines 38-40: This paragraph needs to be larger with more elaboration about types of hydrogels and what has been found with using these so far, rather than jumping straight into the ‘corticoids’ paragraph.

As suggested by the reviewer, the paragraph line 38-40 is completed. The following paragraph is added line 44 : “An enhanced residence time will increase the time over which absorption can occur and the total amount of drug absorbed and has been shown to result in prolonged effect and increased bioavailability in several studies [5]. As example, this concept is used in marketed eye drop such as TIMOPTOL LP® and GELTIM LP which are instilled once daily vs twice daily with TIMOCOMOD®.”

4-       Line 53 The sentence does not do a good job justifying the use of DXMa in the study

The sentence: « Although lipophilic and practically insoluble in water (0,021 mM at pH = 6.0 at 25°C) [9], DXMa (Fig. 1) was selected» is removed from the text and replaced by « Therefore, DXMa (Fig.1) was selected in this study to be formulated for topical ocular administration ».

5-       Figure 1: the labelling of “rings” looks messy. The authors are requested to revise the style of arrows used.

As suggested, the figure 1 is modified in the revised version.

6-        Reference 14 has been unnecessarily placed within the methods. Reference 23 in the results and discussion section also appears to be unnecessarily placed.

References 14 (line 91) and 23 (line 230) are removed in the revised manuscript.

7-       Please substitute “well-known” with “established”.

The term established is used in the text (lines 140 and 178).

8-         Figure 2: Specify within the figure what is in the beaker/container during step 1 (i.e. the complex).

 As suggested, the Figure 2 is modified and completed.

9-           Line 161: Why is Table 6 mentioned before tables 3-5?

 Table 6 is mentioned after table 5.

10-     Equations are interspersed between both the ‘methods’ and ‘results and discussion’ sections. The authors are advised to keep ALL equations within the ‘methods’ section.

Equations 7 and 8 have been moved to the methods section.

11-     Table 9 and Figure 7: is the DXMa content consistent in all the gels tested? Can gels with unequal drug concentrations be compared?

The drug concentrations of the two gels are different. However, the amount of DXMa in the cell extraction is comparable for both gels since 1.5g of gel A (0.7% DXMa) and 0.5g of gel B (2% DXMa) was used respectively. Nevertheless, we agree with the reviewer that the comparison between gels A and B is considered as relative.

Round  2

Reviewer 2 Report

N/A

Author Response

No response

Reviewer 3 Report

The abstract is confusing "retention time" and "release". The authors wrote that the objective was to increase 'retention' of the formulation, but at no point during the manuscript did they test retention. Release on the other hand was studied; please adjust accordingly and more explicitly advertise that this is a complexation study more so than it is an ocular drug delivery study.

The introduction also needs to be reworked. "Ocular inflammation" is not an eye disorder; it is the consequence of many potential ocular disorders. The authors need to revamp the introduction section and focus on a particular disease e.g. discuss specifically uveitis and address the epidemiology of this disease.

Figure 2 caption - please specify which experimental design this figure relates to i.e. was this the method that the authors used in order to prepare the final gel? Or was this just the method used for optimisation studies? The reviewer does not understand the rationale for adding a large excess of DXMa; is this performed just during optimisation studies or is this step included when formulating the actual gels?

Author Response

The abstract is confusing "retention time" and "release". The authors wrote that the objective was to increase 'retention' of the formulation, but at no point during the manuscript did they test retention. Release on the other hand was studied; please adjust accordingly and more explicitly advertise that this is a complexation study more so than it is an ocular drug delivery study.

Response: As asked by the reviewer, the abstract was modified:

“Dexamethasone acetate (DXMa) has proven its efficiency to treat corneal inflammation, without a great propensity to increase intraocular pressure. Unfortunately, its poor aqueous solubility associated with a rapid precorneal elimination result in a low drug bioavailability and a low penetration after topical ocular administration. Themain objectivesof this studywere two fold, improving  was to improve the apparent aqueous solubilityand the ocular residence timeof DXMa using cyclodextrins. First, hydroxypropyl-β-CD (HPβCD) and hydroxypropyl-γ-CD (HPγCD) were used to enhance DXMa concentration in aqueous solution. The β and γ HPCD derivatives allowed to increase the DXMa amount in solution at 25°C by a factor of 500 and 1500 respectively. Second,with the aim to improving the persistence of the complex solution after instillation in the eye, the formulationsof DXMa based-CD solutions with marketed ophthalmic gels (CELLUVISC®, GEL-LARMES® and VISMED®) were investigated and optimized by means of special cubic mixture designs, allowing to define mixed gels loaded with 0.7% (HPβCD) and 2% (HPγCD) DXMa with osmolality within acceptable physiological range.Finally, Iin vitro drug release assays from the mixed gels were performed and compared with reference eye drops.Similarly to MAXIDEX® and DEXAFREE®, in the case of mixed gel containing HPβCD, more than 90% of the drug was released within 2 hours while in mixed gel containing HPγCD, the release of DXMa was partial reaching ≈ 60% in 2 hours. This difference will have to be further addressed with ex vivo andin vivoocular deliveryexperiments.”

The introduction also needs to be reworked. "Ocular inflammation" is not an eye disorder; it is the consequence of many potential ocular disorders. The authors need to revamp the introduction section and focus on a particular disease e.g. discuss specifically uveitis and address the epidemiology of this disease.

Response :As asked by the reviewer, the introduction was modified:

“Ocular inflammation is a common eye disorder is the consequence of many potential eye disorders among which uveitis, is believed to be the cause of about 10% of the cases of severe visual handicap in the United States(1).and anumber of reports demonstrate that ocular administration of anti-inflammatory drug, steroidal (SAID) and non-steroidal (NSAID) drugs are effective in treating ocular surface and anterior segment inflammation[1].

Topical administration of anti-inflammatory drug, steroidal (SAID) and non-steroidal (NSAID), is the most frequently used method to treat ocular surface and anterior segment inflammation as it presents an easy accessibility, a simplicity of use, a non-invasive way and generally a good tolerance. Nevertheless, the ocular drug bioavailability in conventional eye drops is notoriously poor, only 1-5% of drug applied to the surface penetrates the cornea. This is the consequence of various effective protective mechanisms and multiple barriers to drug entry, including a fast naso-lachrymal drainage due to high tear fluid turnover and lid blinking, the corneal structure with a hydrophilic stroma sandwiched between the lipophilic epithelium and endothelium, epithelial drug transport barriers and clearance from the vasculature in the conjunctiva [2,3].

Numerous strategies have been developed to increase the bioavailability of ophthalmic drugs. One of them is to prolong the contact time between the drug and the corneal/conjunctival epithelium by the use of mucoadhesive hydrogels [4]. Anenhancedresidence time will increase the time over which absorption can occur and the total amount of drug absorbed and has been shown to result in prolonged effect and increased bioavailability in several studies[5]. As example,thisstrategyisused in marketed eye drop such as TIMOPTOL LP® and GELTIM LP whichare instilled once ddaily vs twice daily with TIMOCOMOD®.

Among corticoids, dexamethasone (DXM) is having one of the highest potency and effectiveness on inflammation.DXM is used forthetreatment of acute and chronic eyeinflammation, including post-operative inflammation or uveitis[6,7]. DXM acts in the human trabecular meshwork cells by inhibiting phospholipase-A2, i.e. prostaglandins synthesis which causes inflammation [8,9]. Unfortunately, DXM is presenting a formulation challenge, since it is a water insoluble compound [10,11]. DXM is soluble to a limited extent in aqueous eye drop. Thus the drug is frequently used as suspensions, such as 0.1% w/v MAXIDEX® (Novartis Pharma) or as solutions using a hydrophilic water-soluble prodrug, such as 1% w/v DXM sodium phosphate DEXAFREE® (Laboratoires Théa) [7]. Thelipophilic derivative DXM acetate (DXMa), currently unavailable for topical use has been shown to readily permeate the cornea and hydrolyze to DXM during absorption [12]. As well, Leibowitz et al. demonstrated that the acetate form was more effective compared to phosphate derivative in suppressing inflammation in the cornea. This therapeutic effect was not associated to a greater propensity to increase intraocular pressure, one of the most frequent side effects of glucocorticoids [13]. Although lipophilic and practically insoluble in water (0,021 mM at pH = 6.0 at 25°C) [9],DXMa(Fig. 1) was selected in this study to be formulated for topical ocular administration.Therefore, DXMa(Fig. 1)was selected in this study to be formulated for topical ocular administration. Methods such as pH adjustment, cosolvency, micellization, complexation or use of cyclodextrins (CD) are among the most commonly used approaches for drug solubilization allowing the formulation of eye drops solution [14]. Cyclodextrins present the great advantage to enhance both bioavailability and the apparent solubility of poorly water soluble drug while being biocompatible [15,16]. In this context, the two hydrophilic cyclodextrin derivatives hydroxypropyl-β-cyclodextrin (HPβCD) and hydroxypropyl-γ-cyclodextrin (HPγCD) were used to enhance the molecular DXMa fraction in aqueous solution. The mixture of CD/DXMa solutions with marketed mucoadhesive gels were investigated as topical drug vehicles to the eye, with the objectives to achieve therapeutically effective DXMa dosage form with a reduced frequency of instillation.”

Figure 2 caption - please specify which experimental design this figure relates to i.e. was this the method that the authors used in order to prepare the final gel? Or was this just the method used for optimization studies? The reviewer does not understand the rationale for adding a large excess of DXMa; is this performed just during optimization studies or is this step included when formulating the actual gels?

Response: Solubility experiments have to be performed with drug saturated solutions in order to evaluate the ability of the system to maximize the solubilized drug. Moreover, in the case of cyclodextrins, the formation of inclusion compound is favored when using a large excess of the guest molecules. So, the step 2 (Figure 2, addition of an excess of drug) was performed during the 29 experiments of special cubic mixture designs and during optimization studies. Then, the predictive models of the experimental designs provide the exact amounts of the three components (DXMa, CDs, hydrogel(s)) to be used in gels A and B. In this case, no drug in excess was used in order to avoid precipitation risk of DXMa.